# Blockade of Glycosphingolipid Synthesis Inhibits Cell Cycle and Spheroid Growth of Colon Cancer Cells In Vitro and Experimental Colon Cancer Incidence In Vivo

**DOI:** 10.3390/ijms221910539

**Published:** 2021-09-29

**Authors:** Richard Jennemann, Martina Volz, Felix Bestvater, Claudia Schmidt, Karsten Richter, Sylvia Kaden, Johannes Müthing, Hermann-Josef Gröne, Roger Sandhoff

**Affiliations:** 1Lipid Pathobiochemistry Group, German Cancer Research Center, 69120 Heidelberg, Germany; m.volz@dffz.de (M.V.); r.sandhoff@dkfz.de (R.S.); 2Light Microscopy Facility, German Cancer Research Center, 69120 Heidelberg, Germany; f.bestvater@dkfz.de (F.B.); c.schmidt@dkfz.de (C.S.); 3Core Facility Electron Microscopy, German Cancer Research Center, 69120 Heidelberg, Germany; k.richter@dkfz.de (K.R.); s.kaden@dkfz.de (S.K.); 4Institute for Hygiene, University of Münster, 48149 Münster, Germany; jm@uni-muenster.de; 5Medical Faculty, University of Heidelberg, 69120 Heidelberg, Germany; groene@staff.uni-marburg.de; 6Institute of Pharmacology, University of Marburg, 35043 Marburg, Germany

**Keywords:** glucosylceramide synthase, glycosphingolipids, azoxymethane, dextrane sulfate, colorectal cancer, cationic amphiphilic drugs

## Abstract

Colorectal cancer (CRC) is one of the most frequently diagnosed cancers in humans. At early stages CRC is treated by surgery and at advanced stages combined with chemotherapy. We examined here the potential effect of glucosylceramide synthase (GCS)-inhibition on CRC biology. GCS is the rate-limiting enzyme in the glycosphingolipid (GSL)-biosynthesis pathway and overexpressed in many human tumors. We suppressed GSL-biosynthesis using the GCS inhibitor Genz-123346 (Genz), NB-DNJ (Miglustat) or by genetic targeting of the GCS-encoding gene UDP-glucose-ceramide-glucosyltransferase- (*UGCG*). GCS-inhibition or GSL-depletion led to a marked arrest of the cell cycle in Lovo cells. *UGCG* silencing strongly also inhibited tumor spheroid growth in Lovo cells and moderately in HCT116 cells. MS/MS analysis demonstrated markedly elevated levels of sphingomyelin (SM) and phosphatidylcholine (PC) that occurred in a Genz-concentration dependent manner. Ultrastructural analysis of Genz-treated cells indicated multi-lamellar lipid storage in vesicular compartments. In mice, Genz lowered the incidence of experimentally induced colorectal tumors and in particular the growth of colorectal adenomas. These results highlight the potential for GCS-based inhibition in the treatment of CRC.

## 1. Introduction

Colorectal cancer (CRC) is one of the most commonly diagnosed malignant cancers and the most common cancer of the gastrointestinal tract in humans [1]. Intestinal adenomas are benign polyps of the colon mucosa with the potential to mutate into cancerous lesions [2]. Certain risk factors foster the emergence of colorectal carcinomas. The majority of CRC occur in patients over 50 years of age. Excessive consumption of alcohol, nicotine abuse, consumption of red meat, and obesity are each associated with the prevalence of colorectal tumors [3,4]. Chronic colonic inflammation such as ulcerative colitis has also been linked to the development of cancer. Approximately 30% of CRC exhibit increased familial risk, due to inherited genetic polymorphisms [5].

Surgery is the treatment of choice for CRC at stages I-II and is paired with chemotherapy for the treatment of stages III-IV [6]. However, chemotherapy is frequently accompanied by drug resistance limiting its therapeutic success [7].

In search for an alternative or additional treatment strategy, we focused on the potential impact of a reduction of glycosphingolipid (GSL)-synthesis on CRC biology using in vitro and in vivo models. In a previous study we showed that the incidence of chemically induced hepatocellular carcinoma was markedly inhibited in mice after genetic deletion of hepatocyte specific UDP-glucose-ceramide-glucosyltransferase (*Ugcg*) [8]. 

The basic GSL, glucosylceramide (GlcCer), is synthesized from UDP-glucose and ceramide (Cer) at the cytosolic surface of the Golgi apparatus [9]. Glucosylceramide synthase (GCS) is encoded by the gene *Ugcg* in rodents and *UGCG* in humans and represents the rate limiting enzyme catalyzing the first step in the GSL-biosynthesis pathway (Figure 1A). GlcCer can be elongated after transition to the Golgi lumen by the addition of additional carbohydrates leading toward complex GSL-series and gangliosides (Figure 1A). GSLs are then shuttled to the plasma membrane where they are involved in processes such as cell adhesion, membrane receptor activation, signaling and endocytosis [10,11]. Interference in these cellular processes can influence cell proliferation, differentiation, migration, and survival. Silencing of GCS may thus interfere with cancer cell growth as shown in in vitro cell culture experiments and xenograft cancer models [12,13,14,15,16,17]. 

We show here that that inhibition of GSL-synthesis with *N*-[(1*R*,2*R*)-1-(2,3-dihy- dro-1,4-benzodioxin-6-yl)-1-hydroxy-3-pyrrolidin-1-ylpropan-2-yl]nonanamide (Genz- 123346 or Genz), n-butyl-deoxynojirimycin (NB-DNJ) (Miglustat) and by targeting *UGCG* with specific guide RNA by Crispr/Cas9 technology suppressed cell proliferation and tumor spheroid growth in Lovo and HCT116 human colon carcinoma cells in vitro. Using an experimental mouse cancer model mimicking many features of human CRC [18], tumor incidence in Genz-treated mice was shown to be significantly lower than in non-treated controls.

## 2. Results

### 2.1. Gene Expression Analysis of UGCG in Human Colorectal Adenocarcinomas

GCS the basic enzyme in the GSL-biosynthesis pathway (Figure 1A) is overexpressed in multiple human cancers including hepatocellular carcinoma, breast, cervix, non-small-cell lung cancer, and papillary thyroid carcinoma [8,19,20,21], and also in many cancer cell lines [22]. *UGCG* expression data from human colorectal adenocarcinomas suggested a lower expression in tumor as compared to normal colon tissue (Figure 1B). Importantly, the overall survival time of patients with high *UGCG* expression in tumors decreased significantly as compared to patients with low *UGCG* expression (Figure 1C) suggesting a potential impact of *UGCG* expression on cancer prognosis and patient survival. Data were evaluated using the gene expression profiling interactive analysis tool GEPIA (http://gepia.cancer-pku.cn/index.html accessed on 16 December 2020).

### 2.2. Treatment of Colon Carcinoma Cells with the GCS Inhibitor Genz Leads to Depletion of GSLs and to an Arrest of the Cell Cycle

We investigated the effects of lowering sphingolipid synthesis using the GCS inhibitor Genz (Figure 2A) on human Lovo- and HCT116 colon-carcinoma cell lines on cell proliferation. Both cell lines expressed predominantly neutral GSLs but their patterns differed markedly (Figure 2B,C). Untreated Lovo cells lacked globo-series GSLs (Figure 2D) and as judged from their migration on thin layer chromatography (TLC)-plates (Figure 2B). MS^2^ analysis further showed expression of GSLs of the lacto-series, i.e., Lc_4_Cer, Lc_5_Cer and Lewis^a^ (fucosylated Lc_4_Cer, (Appendix A)). (GSLs are abbreviated according to the joint commission on biochemical nomenclature of glycolipids [23]). HCT116 cells in contrast expressed predominantly Gb_3_Cer and Gb_4_Cer, GSLs of the globo-series (Figure 2C). HCT116 cells showed low content of acidic GSLs, principally co-migrating with the gangliosides GM3 and GD3 (Figure 2C). GSLs significantly decreased in both cell lines following treatment with 1 and 5 µM Genz seen after four days (Figure 2B,C and Appendix A). A residual amount of neutral GSLs migrating on the level of HexCer remained in both cell lines likely consisting of GalCer whose synthesis is not effected by GCS inhibition (see also Appendix A). Treatment of Lovo cells with 1 µM and 5 µM Genz caused an arrest in the cell cycle (Figure 2E,F, quantification). An elevated number of Genz-treated cells arrested in G0/G1 phase whereas cell numbers in S and G2/M phase decreased as compared to untreated control cells. Only Genz-concentrations of 5 µM or above markedly inhibited the cell cycle of HCT116 cells (Figure 2G,H, quantification). Genz treatment did not alter cell viability of both cell lines (Appendix A). 

Genz treated cells were quantified after 4 days of culture and effects were further investigated using three-dimensional tumor spheroid cultures. The number of Lovo and HCT116 cells markedly decreased upon treatment (Appendix A). Treatment with Genz also led to a reduction in tumor microsphere size of both Lovo- (Figure 2I,J, quantification) and HCT116 cells (Figure 2K,L, quantification). 

### 2.3. UGCG-Inhibition by NB-DNJ (Miglustat) Also Affects Cell Cycle and Tumor Spheroid Growth of Lovo and HCT116 Cells

The iminosugar Miglustat (Figure 3A) at a concentration of 100 µM led to an almost complete GSL-depletion in Lovo (Figure 3B) and HCT116 cells (Figure 3C) after four days of culture.

Miglustat inhibited the cell cycle of Lovo cells in a manner similar to that seen with Genz (Figure 3D,E, quantification) but did not appear to impair the cell cycle of HCT116 cells (Figure 3F,G, quantification). Miglustat-treatment markedly inhibited tumor spheroid growth by both Lovo- (Figure 3H,I, quantification) and HCT116 cells (Figure 3J,K, quantification). The effects were small but significant after 6 days of culture but resulted in a highly relevant reduced sphere growth after 9 days of culture. 

### 2.4. UGCG-Gene Deletion by Crispr/Cas9 Technology

Lovo- and HCT116 cells were targeted with *UGCG*-guide-RNA (gRNA) to help validate whether the cell cycle arrest seen, particularly observed in Lovo cells, could be assigned to the GCS-inhibition with Genz and Miglustat or occurred due to a secondary effect of the drugs. GSLs were successfully depleted in both cell lines (Figure 4A,D).

A double band in the extracts of both cell lines migrating at the height of HexCer remained in both *UGCG*-gRNA-treated cell lines. A perborate separation of GlcCer from GalCer on TLC indicated the presence of GalCer (Appendix A). An additional double band remained in *UGCG*-depleted HCT116 cells migrating at the height of lactosylceramide (LacCer). This compound likely consists of galabiosylceramide, which in similar manner as seen with GalCer was not targeted by *UGCG*-gRNA. We further investigated whether genetic GSL-depletion would lead to a comparable arrest of the cell cycle as seen with inhibitor treatment. We could not detect increased numbers of cells in GO/G1 phase or a reduction of cells in the S or G2/M phase of the cell cycle in *UGCG*-depleted HCT116 cells (Figure 4B,C, quantification). This finding is in general agreement with the data obtained from the 1 µM Genz and Miglustat treatment (Figure 2G,H and Figure 3F,G). Lovo cells in contrast showed a moderate arrest of the cell cycle upon *UGCG* depletion (Figure 4E,G, quantification) with elevated numbers of cells seen in GO/G1 phase and lower numbers in S- and G2/M phase. Inhibition of the cell cycle appeared slightly stronger in cells treated with 5 µM Genz (Figure 4F,G, quantification) than seen in the genetically modified cells, suggesting an additional Genz-associated contribution. 

### 2.5. Treatment of Lovo and HCT116 Cells with Genz Causes Lipid Accumulation in Multivesicular Bodies (MVBs)

Ultrastructural investigation of Genz-treated cells by electron microscopy (EM) showed a dose-dependent increase in numbers of MVBs. 

In this respect, the sensitivity of Lovo-cells appeared much higher as compared to HCT116-cells: While the increase in MVB-numbers was seen in Lovo-cultures at low 1 µM Genz treatment, HCT116-cells required higher doses (Figure 5A). The ultrastructure of membrane-organization also differed (Figure 5B,C). MVBs in HCT116-cells display strange corner shapes (Figure 5C). In both cell-lines vesicles inside MVBs exhibit multilamellar composition with potential of membranes to anneal into close stacks (Figure 5C and Appendix A). The number of abnormal structured multilamellar and lipid rich MVBs rose with increased Genz concentrations (Figure 5D). Our results are in agreement with a recent paper reporting elevated sphingomyelin storage in lysosomal compartments of cells treated with the GCS inhibitor PDMP [24]. The periodic repeat in those areas accounts for a membrane thickness of 3.8 to 4 nm. Analysis at optical definition to resolve the bi-layer of membranes suggested the presence of blunt ends, micellar convolutions as well as widening of the expectedly lipophilic inner leaflet suggesting perturbation of canonical unit-membrane organization (Appendix A). The alterations in vesicles detected by electron microscopy coincided with increased expression of the autophagy marker Lc3-II at elevated Genz-concentrations in both Lovo and HCT116 cells (Figure 5E,F).

### 2.6. GSL-Synthesis Inhibition by Miglustat or Ugcg-gRNA Did Not Cause Intra-Lysosomal Lipid Accumulation

In contrast to the Genz-treated cells, a lysosomal accumulation of lipids was not observed upon Miglustat- (Appendix A) or *Ugcg*-gRNA treatment (Appendix A) of Lovo- (Appendix A) and HCT116 cells (Appendix A). 

### 2.7. MS^2^ Analysis of Colon Carcinoma Cells Indicated a Marked Sphingomyelin Increase upon GCS-Inhibitor Treatment

Mass-spec analysis of Genz-, Miglustat- and *UGCG*-gRNA- treated Lovo- and HCT116 cells was performed to identify specific effects on sphingolipid synthesis. The levels of HexCer in Lovo and HCT116 cells decreased to a similar extent independent of the method of GCS-silencing (Figure 6A–D). Sphingomyelin (SM)- and phosphatidylcholine (PC) content in Genz-treated Lovo (Figure 6A) and HCT116 cells (Figure 6B) increased as compared to controls. The SM and PC content rose with elevated Genz concentrations and was much higher as in gRNA-treated cells. Ceramide levels however did not appreciably change and were even slightly lower in *UGCG*-gRNA treated cells as compared to controls (Figure 6A,B). 

Lovo- and HCT116 cells treated with Miglustat also showed elevated SM content as compared to controls (Figure 6C,D). The absolute increase was not as prominent as that seen in cells treated with Genz. PC increased only slightly in Lovo cells but not in HCT116 cells (Figure 6C). Miglustat application did not alter ceramide content of Lovo and HCT116 cells (Figure 6C,D).

### 2.8. Genz Treatment of Mice Lowers Incidence of Experimental Induced CRC

We then sought to validate these in vitro findings using a mouse model of experimental CRC. Colorectal tumors were induced with azoxymethane (AOM) in mice at the age of 8 to 10 weeks (Figure 7A). One week later, mice received dextrane sulfate (DSS) for five consecutive days. Mice received food either with or without Genz supplementation five weeks after tumor induction (Figure 7A). All animals were sacrificed fifteen weeks after CRC induction and tumor load and size were evaluated. Human colorectal tumors predominantly occur in regions of the sigmoid colon and the rectum [25,26] (Figure 7B). In the AOM/DSS model used here, Genz- and chow-fed animals developed tumors particularly in the rectal part of the colon (Figure 7B). Both number and size of the tumors treated with Genz were reduced (Figure 7C). The overall tumor number following Genz treatment was significantly lower as compared to control mice (5.10 ± 0.99 vs. 10.29 ± 1.29; *p* < 0.01). A reduction was observed for tumors smaller and larger than 2 mm in diameter (2.80 ± 0.51 vs. 5.00 ± 0.59 for tumors < 2 mm; *p* < 0.05 and 2.30 ± 0.70 vs. 5.29 ± 0.99 for tumors > 2 mm; *p* < 0.05). The mean tumor diameter of all tumors per mouse was smaller in the Genz-treated cohort as compared to tumors grown in colons of control mice (1.7 ± 0.2 mm vs. 2.1 ± 0.1 mm; *p* < 0.05) (Figure 7D). A total of 80.7% of the tumors from the Genz-treated mice and 89.1% of the tumors that developed in chow fed animals localized at the distal part of the colon, preferentially in the rectum and close to the anus (Figure 7B,C). A lower number of tumors grew in the medial part (colon transversum) of the colons of both cohorts. Tumors were almost absent in the proximal part (colon ascendum). All mice developed predominantly colorectal adenomas with markedly lower numbers upon Genz-feeding (Figure 8A,B). Ki67-positive cells were determined to elucidate a potential effect of Genz treatment on the proliferation of cells in crypts. Control mice contained a mean of 19.6 ± 0.24 and Genz fed animals 17.8 ± 0.22 (*p* < 0.001) Ki67-positive cells in tumor-free colon crypts (Figure 8C) suggesting an effect on cell division of highly proliferative cells.

Immunohistochemistry of the colon sections for F4/80+ cells indicated the presence of macrophages preferentially in the periphery of the tumor of both cohorts. Limited macrophages were seen to penetrate into the tumor (Figure 8D, left panel). Differences between Genz-treated mice and controls were not detectable. Very low numbers of T-cells could be detected with anti-CD3 antibodies (Figure 8D, middle panel). The number of TUNEL-positive, apoptotic cells in tumors did not differ between Genz-treated mice and controls (Figure 8D, right panel). The majority of these cells contained smaller nuclei and therefore could be distinguished from epithelial cells.

### 2.9. GSLs Are Reduced and SM Levels Elevated in the Intestine upon Genz-Feeding 

Sphingolipid analysis by TLC and MS^2^ was then performed to elucidate potential alterations in sphingolipid content due to the Genz treatment of the mice. GlcCer and Gg_4_Cer (GA1) were the two major components detected in the intestine of adult mice (Figure 9A) [27]. The ratio between HexCer and GA1 was determined to be approximately 1:0.4. GM3 was only found within the acidic GSL-fraction in low concentrations (Figure 9A). MS^2^ analysis showed a reduction, particularly of the NS-, AS-, and to a lesser extent, AP-containing GSL-species in Genz-treated mice (Figure 9B,C). Although NS-, AS- and AP-species GSLs decreased, surprisingly, NP-Cer-levels were found to be markedly elevated (Figure 9D). Within SM, the NS-species increased upon Genz feeding (Figure 9E). TLC-analysis from heart, spleen, kidney, and liver indicated a strong decrease of GSLs in those organs upon Genz treatment (Appendix A, quantification). Genz-feeding of mice only magically influenced their blood chemistry as compared to the chow cohort (Appendix A).

## 3. Discussion 

To evaluate an alternative treatment of CRC we investigated the impact of GCS-silencing on cell cycle and tumor spheroid growth in vitro in Lovo and HCT116 human colon carcinoma cells. The potential of the GCS inhibitor Genz-123346 in an endogenous animal model of CRC was further validated in vivo. Lovo cells showed cell cycle arrest upon GCS-inhibition, which was similar in both the genetically modified- and inhibitor-treated cells suggesting an impact of GSLs on cell division. GSL-synthesis inhibition did not appear to strongly affect the cell cycle in HCT116 cells. Although 1 µM Genz in the culture medium led to almost a complete depletion of GSLs in treated cells, the cell cycle was more inhibited at elevated doses of 5 µM Genz and higher. Increased Genz concentrations suggested additional effects beyond silencing of GCS in HCT116 cells. In order to investigate the effect of Genz on CRC cells over a longer time, we monitored the growth of tumor microspheres in a three dimensional system. Both HCT116 and Lovo cells showed decreased tumor microsphere growth with pronounced growth retardation of Lovo cell spheroids at low concentrations of Genz or Miglustat.

Our results obtained from the GCS-inhibition in colon carcinoma cells were in agreement with findings from other malignant cancer cells in vitro. A series of studies have suggested that GCS silencing with GCS inhibitors could potentially alter the growth of melanoma [16,28], brain cancer [17,29], lung carcinoma [30], myeloid leukemia [14], hepatocellular carcinoma [8], and myeloma cells [31]. In these studies, either synthetic ceramide analogs such as d-threo-1-phenyl-2-decanoylamino-3-morpholino-1-propanol (PDMP) and (D,L)-threo-1-phenyl-2-hexadecanoylamino-3-pyrrolidino-1-propanol (PPPP), or imino sugars such as Miglustat (NB-DNJ), and *N*-pentyl deoxyidonojirimycin (NP-DIJ; OGT2378) were used to inhibit GCS.

GCS inhibition has been associated with an accumulation of ceramides. Ceramides may foster apoptosis [32] and thereby affect tumor growth. Markedly elevated ceramide levels, however, were not observed here either the Lovo or HCT116 cells upon inhibitor treatment. Therefore, ceramides can be largely excluded as candidates for the cell cycle inhibition and tumor microsphere growth seen in these cells. A marked elevation of SM and PC in was observed in Lovo cells treated with Miglustat which was higher than that seen in either Lovo or HCT116 cells upon Genz treatment. The levels of SM and PC, however, far exceeded the levels of those lipids found in *UGCG*-gRNA-treated cells implying the presence of additional targets of the inhibitors, in particular of Genz, in the context of sphingolipid metabolism. As shown by electron microscopy, lipids in Genz-treated cells accumulated in lysosomes in multilayered, partly compacted- onion-like membrane fashion. 

Normally sphingolipids and phospholipids are enzymatically degraded in lysosomal compartments. Acidic pH conditions are required for this process in order to avoid accumulation. Reports suggest that cationic amphiphilic drugs (CADs) may interfere with a regular catabolism of those lipids leading to severe lipid storage in lamellar structures in endolysosomes [33,34]. CADs are small molecules consisting of a hydrophobic tail with one or more aromatic rings and a hydrophilic amine, which can be protonated in lysosomes. Positively charged CADs interfere with the negatively charge found in intra-lysosomal vesicles required for lysosomal enzyme stabilization and regular lipid degradation [33,34]. Thereby, CADs are thought to influence the catalytic degradation of lipids such as SM by acidic sphingomyelinase, which needs an acidic milieu for proper activity [34,35]. The GCS inhibitor Genz fulfills the general structural properties of such a drug (Figure 2A) and may thus help explain the elevated lipid storage seen in lysosomes. This effect may help contribute to the GSL-depletion seen and influence the cell cycle. A similar accumulation of SM had recently been reported for HeLa cells treated with the GCS inhibitor PDMP [24] that displays similar relevant structural features as seen in Genz. Epidemiologic studies have highlighted a positive anti-cancer effect of CADs. Patients who had received CAD-based tricyclic antidepressants and -antihistamines before cancer diagnosis showed improved survival [36,37,38,39,40,41]. In support of these epidemiological studies in humans, antidepressants which selectively inhibit the reuptake of serotonin (SSRIs) such as citalopram, escitalopram, fluoxetine, and sertraline, which all are considered CADs, have been described in animal models to inhibit the growth of colon, prostate, and lung cancers as well as gliomas and melanomas [34]. Interestingly, CADs have also been described to induce a reversal of the multidrug resistance (MDR) as demonstrated in various murine cancer models [34]. MDR is one of the major pitfalls that occurs during chemotherapeutic treatment of CRC [7]. GCS elevation in cancer has been correlated with drug resistance [22,42,43,44]. GCS up- or down-regulation has been shown to influence expression of the multidrug resistance protein 1 (MDRI) that fosters resistance to many chemical drugs [45,46,47]. Thus, lowering of GCS could potentially enhance drug sensitivity in tumors [48,49,50] and thus help overcome chemoresistance [51,52,53,54,55,56]. GCS inhibitors such as Genz can substantially interfere with GSL-synthesis and through CAD-actions may help limit MDR.

Building upon our in vitro data and previous animal studies with selective CADs, we investigated the impact of the GCS inhibitor Genz on CRC growth in a spontaneous CRC mouse model. The chemical agent AOM was used to initiate colon cancer. AOM induces DNA alkylation which facilitates genomic mutations leading to cancer [57]. Mice additionally received DSS, which promoted cancer development by inducing colitis. Animals of the two cohorts, chow fed and Genz-treated mice developed particularly adenomas. However, we found a marked reduced incidence and smaller tumors in mice treated with the GCS inhibitor and cationic amphiphilic drug Genz. We detected reduced GSL- and elevated SM-levels in the intestine of Genz-treated animals, but the changes were not as much pronounced as in the in vitro cell culture studies. However, a reduced number of Ki67-positive cells in the crypts of mouse colons indicated an effect of Genz on proliferative cells. The differences between treated and non-treated mice were faint, but distinct, and could help explain the reduced tumor numbers and size of tumors that developed over an extended period of time. Importantly, GCS inhibitors have been used successfully in preclinical studies to reduce substrate in GSL-storage disease patients such as Fabry and Gaucher disease [58,59]. 

Although the exact molecular mechanisms of the Genz-dependent decrease of the tumor incidence in the CRC-model need to be further elucidated, the use of GCS inhibitors in the treatment of human colorectal cancer may represent a novel therapeutic approach.

## 4. Materials and Methods

### 4.1. Validation of the UGCG Expression in Human CRC and UGCG-Related Survival 

*UGCG* gene expression analysis of human colorectal adenocarcinomas and survival time of patients with high and low *UGCG* expression in tumors was performed using the TCGA normal and GTEx database in the gene expression profiling interactive analysis tool GEPIA (http://gepia.cancer-pku.cn/index.html, accessed on 16 December 2020).

### 4.2. GCS Silencing of Lovo and HCT116 Cells with Chemical Inhibitors

Lovo and HCT116 cells (ATCC, Manassas, VA, USA) were cultivated in MEM-alpha medium (Sigma, Munich, Germany), supplemented with 10% FCS, 2 mM L-glutamine, and 50 units/mL each of Pen/Strep (all from Life Technologies, Darmstadt, Germany) at 37 °C and 5% CO_2_ atmosphere. 2 × 10^5^ cells each were cultivated in 10 mL culture medium in- or without the presence of the indicated concentrations of Genz-123346 (*N*-[(1*R*,2*R*)-1-(2,3-dihydro-1,4-benzodioxin-6-yl)-1-hydroxy-3-pyrrolidin-1-ylpropan-2-yl]nonanamide) (Chess, Mannheim, Germany) or with 100 µM NB-DNJ (Miglustat, Biozol, Eching, Germany). Cells were cultivated in 10 cm tissue culture dishes (Greiner Bio One, Frickenhausen, Germany) for 4 days. Cells were counted, split and 10^6^ cells treated for another 2 days, each in triplicates. Cells were harvested and used for FACS-, protein-, and TLC-analysis.

### 4.3. UGCG Deletion in Lovo Colon Carcinoma Cells Using CRISPR/Cas9 Technology 

A CRISPR/Cas9 expression vector with puromycin and ampicillin resistance (pX459, Addgene, Cambridge, MA, USA) was used for integration of the human *UGCG*-gRNA. The vector (5 µg) was digested with 10 units of the enzyme Bbs I (Thermo Scientific, Waltham, MA, USA) at 37 °C for 16 h. The DNA was purified using a PCR-purification Kit (Qiagen, Hilden, Germany). 

The guide oligos including the 5′- and 3′-Bbs I overhangs, respectively were:
*UGCG*-guide-119 forward: 5′-caccgcgattacacctcaacaaga-3′; *UGCG*-guide-119 reverse: 5′-aaactcttgttgaggtgtaatcgc-3′.*UGCG*-guide-451 forward: 5′-caccgccatgtcagtaagcgtatc-3′; *UGCG*-guide-451 reverse: 5′-aaacgatacgcttactgacatggc-3′.

Guide oligos, were dissolved at a concentration of 150 µM. 25 µL of the respective forward and reverse oligonucleotides were mixed and annealed (95 °C, 4′; 70 °C, 10′; followed by cooling down the mixture in a thermomixer (Eppendorf, Hamburg, Germany) to room temperature. For oligo insertion, ~100 ng of the purified expression vector, 1 ng of the annealed oligo and 1 unit of T4-ligase and ligase buffer (Life Technologies, Darmstadt, Germany) in a total volume of 20 µL (ad. ddH_2_O) were used. DH5α *E. coli* (Life Technologies, Darmstadt, Germany) were transformed with 5 µL of the ligation mixtures according to the manufacturer’s protocol and spread on LB agar gel plates with 100 µg/mL ampicillin. The plates were incubated at 37 °C for 16 h. Minipreps of single colonies were performed in 5 mL LB medium with 100 µg/mL ampicillin at 37 °C in a bacteria shaker for 16 h. The plasmid DNA was isolated using a plasmid mini kit from Qiagen. Minipreps were checked for correct integration of the *UGCG* oligonucleotides by PCR covering the guide oligo insertion site and consecutive sequence analysis. 

### 4.4. Transfection of Lovo Cells with UGCG-gRNA

2.5 × 10^4^ Lovo cells were seeded in a 24-well plate for 24 h and transfected with a mixture of *UGCG* vector DNA or empty vector as control, P3000 reagent and lipofectamine 3000 (Life Technologies, Darmstadt, Germany) according to the manufacturer’s instructions. The selection of transfected cells with 0.5, 1, and 2 µg puromycin/mL culture medium was started 24 h later. The medium was changed one day later and the cells were cultivated for another 3 days in the presence of puromycin. Cells were expanded thereafter in medium without puromycin. After confluence, cells were split to 6-well plates and then to 10 cm dishes. One aliquot was frozen, and one other analyzed for GSL-depletion as described.

### 4.5. Sphingolipid Extraction of Colon Carcinoma Cells

Cells were harvested after trypsination, washed with PBS and the pellets extracted with 2 mL CHCl_3_/CH_3_OH/H_2_O 10:10:1 by vol. in an ultrasound water bath at 50 °C for 15 min. Supernatants were collected after centrifugation (10 min, 3000 rpm) and the pellet extracted a second time with 2 mL CHCl_3_/CH_3_OH/H_2_O 30:60:8 by vol. as before. Combined supernatants were evaporated. One half of the extracts was dissolved in 1 mL 0.1 M KOH in methanol and saponified at 50 °C for 4 h. The samples were neutralized with glacial acetic acid and evaporated. The sphingolipids were desalted using reversed phase column chromatography (RP18). Therefore, small glass pipettes were filled with 200 µL of RP18 material (Waters Associates, Milford, MA, USA). The columns were preconditioned each with 2 mL methanol, H_2_O, and 0.1 M KCl in H_2_O. The extracts were solved in 1 mL of H_2_O which was then applied to the columns. The reagent tubes were rinsed twice with 0.1 M KCl, which was then also loaded on the columns. Columns were washed with 2 mL H_2_O and lipids eluted with 2 mL methanol. The methanol fraction was dried down. GSLs were separated into neutral and acidic components by ion exchange chromatography. Therefore, glass pipettes were filled with 200 µL of DEAE-Sephadex A25 (GE Healthcare, Chicago, IL, USA), and columns were preconditioned with 2 mL of methanol. The samples were solved in 2 mL of methanol and applied to the columns. Reagent tubes were rinsed two times with 1 mL of methanol, and neutral GSLs were completely eluted from the columns with additional 2 mL of methanol. Acidic GSLs were eluted with 2 mL of 0.5 M potassium acetate in methanol. Acidic GSLs were dried and solved in 5 mL of H_2_O and desalted using preconditioned RP18 columns as described before. An amount corresponding to 0.1 mg of dry intestine was loaded on TLC plates (Merck, Darmstadt, Germany) for the staining of neutral and acidic GSLs. Running solvents are described in the respective Figure Legends. TLC plates were sprayed with 0.2% orcinol in 10% sulfuric acid at 120 °C for 10 min to visualize GSLs.

### 4.6. Anti-Gb_3_Cer Immune Overlay

For the detection of Gb_3_Cer, two GSL-aliquots of Lovo and HCT116 control- and Genz-treated cell extracts, each corresponding to 0.2 mg protein were loaded on a TLC plate (Merck, Darmstadt, Germany). The TLC plate was divided into two parts. One was stained with orcinol reagent as described above to visualize all GSLs. The second plate was fixed with 5% polyisobutylmethacrylate (Sigma, Munich, Germany) in chloroform which was 1:10 diluted in n-hexane. After the organic solvent was evaporated, the plate was blocked with 1% BSA in PBS for 30 min. Blocking solution was removed and the plate subsequently covered with polyclonal chicken anti-Gb_3_Cer antibody 1:200 diluted in 1% BSA in PBS at 4 °C overnight. The plate was washed five times with PBS/0.05% Tween-20 and incubated with secondary 1:500 diluted alkaline phosphatase linked donkey anti-chicken IgY antibodies (Dianova, Hamburg, Germany) in 1% BSA/PBS for 4 h at RT. After washing for five times, positive bands were detected using Sigma Fast BCIP (Sigma, Munich, Germany).

### 4.7. Cell Cycle FACS-Analysis with Propidium Iodide (PI)

Cells were cultivated for six days with or without 1 µM/5 µM Genz as described above. Cells were washed with PBS, trypsinized and transferred to 15 mL conical tubes. The tubes were centrifuged at 1500 rpm and supernatant decanted. The residual amount of liquid was carefully removed with a pipette. The cells were suspended in 250 µL PBS by gently pipetting up and down for approximately five times. Ice cold methanol (700 µL, precooled at −20 °C) was added dropwise while slowly vortexing the tubes. The cells were incubated at 4 °C for at least one hour. Approximately 10^6^ cells were added to 4 mL precooled PBS with 1% FCS into FACS tubes and centrifuged at 1500 rpm for 5 min. Cells were stained with 400 µL PI-solution + RNAse A (0.01 mg/mL PI, 0.25 mg RNAse/mL in PBS/1% FCS). Cells were incubated at RT in the dark for 30 min and immediately measured by FACS in linear mode on a FACS Canto with Diva software (BD, East Rutherford, NJ, USA). Analysis of the data was performed using FlowJo V (Tree Star, Flow Cytometry Analysis Software, Ashland, OR, USA).

### 4.8. Cell Viability Assays

Cells were cultivated in the presence of Genz as described above. The cells were harvested by trypsination, washed with PBS and suspended in PBS/1% FCS, 1 µg/mL PI. The mixture was incubated for 30 min at RT in the dark and subsequently measured by FACS as described. 

### 4.9. Tumor Spheroid Formation of Genz-Treated Cells

Spheroid formation was performed essentially as described [60]. 10,000 cells were placed in each well of a 96-well round bottom ultralow attachment plate (Corning CORN7007, Wiesbaden, Germany) followed by centrifugation at 450× *g* for 5 min. Spheroids were cultured for 24 h in regular medium which then was carefully exchanged by 0-, 1 µM-, 5 µM- and 10 µM Genz-containing medium for 9 days. Half of the medium was replaced every three days containing the respective Genz concentrations. Spheroid size was determined using a microscope (Nikon, Eclipse TS2) and the surface area calculated with ImageJ after 2, 6, and 9 days.

### 4.10. Electron Microscopy of Genz-Treated Cells 

Lovo and HCT116 cells, cultured on aclar-fluoropolymer-sheets (Science Servoces, Munic, Germay) were treated as described with 1 μM or 10 μM Genz for 6 days. Cells were fixed in 2% formic aldehyde/2% glutaric aldehyde/1 mM CaCl_2_/1 mM MgCl_2_/100 mM cacodylate buffer pH 7.2, following post-fixation in 1% OsO4, Uranyl en bloc staining (1% uranyl-acetate in 75% ethanol), dehydration in ethanol and Epon-embedding (Serva, Heidelberg, Germany). Ultrathin sections were cut at 50 nm nominal thickness (UCT, Leica, Wetzlar, Germany), contrasted with uranyl and lead and observed with an EM910 (Carl Zeiss Oberkochen, Germany) equipped with a CCD-Camera (TRS, Morenweis, Germany).

### 4.11. Western Blotting of Genz-Treated Cells

Control- and Genz-treated cells at ~70% confluency were washed with PBS and lysed by pipetting up- and down in digitonin lysis buffer [20 mM HEPES-NaOH buffer (pH 7.4) containing 25 mM KCl, 250 mM sucrose, 2 mM MgCl_2_, with freshly added 1% digitonin (Sigma, Munich, Germany), complete protease inhibitors, phospho-stop (Roche, Basel Switzerland) and 0.5 mM DTT]. Cell lysates were incubated on ice for 30 min and subsequently centrifuged at 13,000 rpm at 4 °C for 10 min. Supernatants were collected and protein content determined with Bradford reagent (Sigma, Munich, Germany). 20 µg of protein was used in a 10% SDS gel. The proteins were blotted on a nitrocellulose membrane (Amersham, GE Healthcare, Chicago, IL, USA) and unspecific binding blocked. The membranes were then incubated with primary rabbit Lc3-I/II antibodies (Cell Signaling #12741, Danvers, MA, USA), 1:1000- or with rabbit anti-Tubulin antibodies (Cell Signaling #2128, Danvers, MA, USA), 1:3000 diluted in 5% BSA/TBS at 4 °C overnight. The blots were washed three times with TBS 0.1% tween 20 and incubated with secondary goat anti-rabbit IgG-HRP (Santa Cruz, Heidelberg, Germany), 1:1000 diluted in 5% BSA/TBS at room temperature for 1 h. After three times washing as above, bands were visualized with ECL solution (RPN2109, Amersham/Merck, Darmstadt, Germany) in a ChemiDoc TM Imaging System (Biorad, Feldkirchen, Germany). Quantification was performed with ImageJ. 

### 4.12. Perborate Separation of GlcCer from GalCer

To distinguish GlcCer from GalCer within the remaining double band in the extracts of cells with GCS-inhibition migrating on the height of HexCer, an aliquot corresponding to 0.2 mg protein was loaded together with standard GSLs on a TLC plate (Merck, Darmstadt, Germany). The plate was sprayed with 1.5% sodiumtetraborate solution in H_2_O, well dried in a vacuum desiccator overnight and then developed in CHCl_3_/CH_3_OH/H_2_O/25% aqueous ammonia, 62.5:30:6:0.5, *v*/*v*. The plate was stained with orcinol reagent as described above to visualize the GSLs.

### 4.13. Animals, Tumor Induction, and Genz-123346-Treatment

C57Bl/6 mice at the age of ~8 weeks received a single dose of 10 mg/kg bodyweight azoxymethane (AOM, Sigma, Munich, Germany) in sterile 0.9% NaCl to initiate CRC [18,61]. The diluted AOM solution was applied i.p. in a volume of 100 µL/10 g bodyweight to each mouse. Furthermore, to promote tumor development, animals received 2% DSS (dextrane sulfate 40, AppliChem, Darmstadt, Germany) in the drinking water for five consecutive days one week after the AOM injection. Mice were fed with a powderized chow diet (# 3437, KLIBA-NAFAG, Kaiseraugst, Switzerland) mashed with an equal amount *w*/*v* of sterilized tapped water for five weeks. Thereafter, one group still received soft chow diet and the second cohort received the same soft food supplemented with 0.225% Genz-123346. Mice were kept in a 12 h day/night cycle in a SPF barrier with environmental enrichment. Institutional veterinarians controlled health of the animals. Animal experiments were approved by federal law (Regierungspräsidium Karlsruhe).

### 4.14. Evaluation of Tumor Incidence

All animals were sacrificed 15 weeks after tumor induction. For evaluation of tumor load and size, the entire colon including the rectum was resected and flushed with ice cold PBS. After longitudinal cut, the colon was spread on a filter paper and immediately immersion fixed with 4% paraformaldehyde in PBS pH 7.4 at 4 °C for 48 h. Tumor numbers in each colon were counted on a stereo microscope. Tumor sizes were determined using a caliper. Formaldehyde fixed colons were divided with a scalpel into three parts (proximal, medial, and distal) and after dehydration with an ascending ethanol gradient subsequently embedded in paraffin for further histological analysis.

### 4.15. Sphingolipid Extraction of Mouse Organs and Mass Spectrometry

One piece of the intestine, liver, kidney, spleen and heart was lyophilized and powderized. 10 mg of each organ was filled into a separate glass vial and adjusted to extraction. Sphingolipid extraction was performed accordingly as described above for the colon carcinoma cells with the difference that the powderized organs were extracted twice with CHCl_3_/CH_3_OH/H_2_O 10:10:1 by volume. 

Isolated sphingolipids including GSLs, Cers, and SMs were investigated by mass spectrometry as described [8,27].

### 4.16. Immunohistochemistry of Colon

The tissue was cut with a microtome (Microm, Leica, Nussloch, Germany), deparaffinized and immunohistochemistry was performed with rabbit anti-Ki67 antibodies (IHC-00375, Bethyl Laboratories Inc., Montgomery, TX, USA) to stain proliferating cells. Biotin labeled anti-rabbit antibodies (1:200, Dianova, Hamburg, Germany) and AP-streptavidin (1:200, Vector, Burlingame, CA, USA) were used to visualize Ki67-positive cells in colons. Macrophages were stained with rat anti-F4/80 (Acris # 4007, Rockville, MD, USA), 1:500 and secondary biotinylated anti-rat/streptavidin-AP (Vector, Burlingame, CA, USA), 1:200. T cells were stained with rat anti-CD3 (#DIA-303, Hamburg, Germany), 1:50 and secondary biotinylated anti-rat (Vector), 1:200 and streptavidin- AP (Vector, Burlingame, CA, USA), 1:200. Images were taken with a Keyence BZ-9000 microscope (Neu-Isenburg, Germany).

### 4.17. TUNEL Assay

TUNEL was performed according to the manufacturer’s instructions of the (In Situ Cell Death Detection Kit, Roche, Mannheim, Germany).

### 4.18. Statistics

Data were analyzed using student’s two-tailed *t*-test. Graphs show mean values ± SEM. Significances are *, *p* ≤ 0.05; **, *p* < 0.01; ***, *p* < 0.001.

## 5. Conclusions

In conclusion, we have shown that glycosphingolipid (GSL)-depletion by inhibition of glucosylceramide synthase, the major enzyme involved in the initial step of GSL-biosynthesis, led to growth retardation of colon cancer cell-derived tumor microspheres and to an arrest of the cell cycle. In a mouse model mimicking human CRC, marked reduction particularly of colorectal adenomas was achieved by treatment of animals with the GCS-inhibitor Genz-123346. The exact molecular mechanism of the Genz-dependent decrease of the tumor incidence in the CRC-model needs to be further elucidated. However, since inhibitors of GSL-biosynthesis are clinically available, this study may have a direct translational implication as a potential therapeutic application for patients suffering from colorectal cancer.

## Figures and Tables

**Figure 1 ijms-22-10539-f001:**
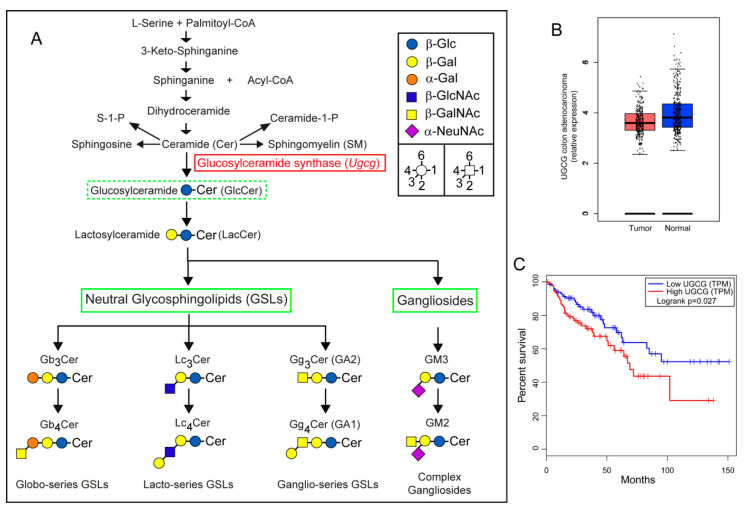
UGCG expression in human colorectal cancer (CRC) with relation to survival. (**A**) Sphingolipid synthesis pathway; the enzyme glucosylceramide synthase (GCS) encoded by the gene *UGCG* is the initial enzyme in the glycosphingolipid (GSL)-biosynthesis pathway. Targeting GCS using genetic approaches or with specific inhibitors such as Genz-123346 disrupts the synthesis of glucosylceramide (GlcCer) and its downstream GSL-products and may lead to accumulation of precursors. (GSLs are abbreviated according to the joint commission on biochemical nomenclature of glycolipids [23]). (**B**) The expression of *UGCG* mRNA was lower in human CRC as compared to normal colon. (**C**) However, patients with high *UGCG* mRNA expression in colorectal tumors had a reduced overall survival time than patients with lower *UGCG* expression.

**Figure 2 ijms-22-10539-f002:**
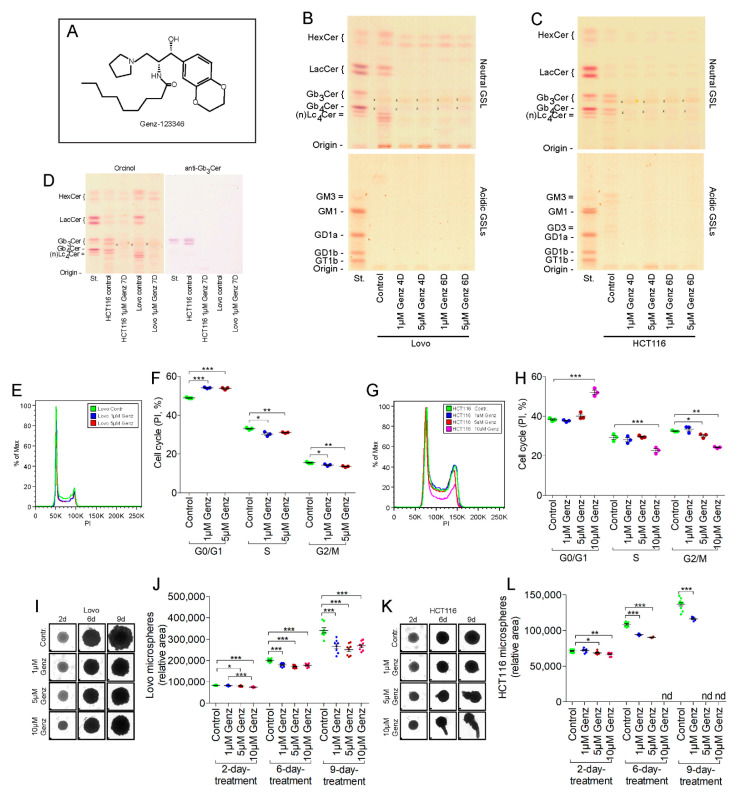
Treatment of Lovo and HCT116 cells with Genz lead to reduced GSL expression and cell cycle arrest. (**A**) Formula of the GCS inhibitor Genz-123346 (*N*-[(1*R*,2*R*)-1-(2,3-dihydro-1,4-benzodioxin-6-yl)-1-hydroxy-3-pyrrolidin-1-ylpropan-2-yl]nonanamide). (**B**,**C**) As shown by thin layer chromatography (TLC), GSL-synthesis was almost completely disrupted by treatment of Lovo (**B**) and HCT116 cells (**C**) with 1 µM Genz in the culture medium after 4 and 6 days; x, no GSL-positive band; running solvent for neutral GSLs was CHCl_3_/CH_3_OH, H_2_O, 62.5:30:6 (*v*/*v*) and for acidic GSLs CHCl_3_/CH_3_OH, 0.2% CaCl_2_, 60:35:8 (*v*/*v*). (**D**) Anti-Gb_3_Cer immune overlay of the neutral GSL-fraction from Lovo and HCT116 cells. Chemical staining with orcinol reagent (left) and immune overlay with anti-Gb_3_Cer antibodies. HCT116 cells expressed globosides in addition to hexosylceramides and lactosylceramides, which were not detectable in Lovo cells. (**E**,**F**, quantification) Cell cycle arrest of the Lovo cells in S and G2/M phases occurred after treatment with 1 µM Genz (**G**,**H**, quantification). A significant influence on the cell cycle of HCT116 cells as shown by FACS analysis occurred at 5 µM Genz and higher concentrations. (**I**–**L**) One micrometer of Genz inhibited growth of tumor spheroids from Lovo (**I**,**J**, quantification) and HCT116 cells (**K**,**L**, quantification). Note: HCT116 microspheres broke apart with elevated Genz concentrations after longer incubation period (**K**) and could therefore not be included in the calculations (**L**, nd, not done); each data point on the graphs represents one biological replicate; scale bars, 100 µM; significances are *, *p* ≤ 0.05; **, *p* < 0.01; ***, *p* < 0.001.

**Figure 3 ijms-22-10539-f003:**
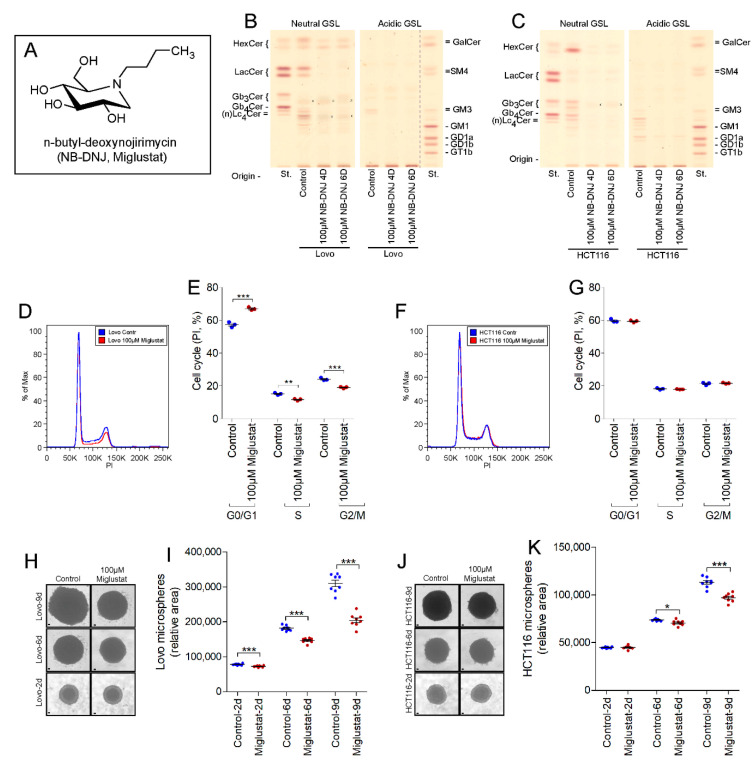
Miglustat treatment leads to cell cycle arrest in Lovo cells and inhibits growth of Lovo and HCT116 tumor spheroids. (**A**) Formula of Miglustat. (**B**,**C**) n-butyl-deoxynojirimycin (NB-DNJ, Miglustat) inhibits GSL-synthesis of Lovo (**B**) and HCT116 cells (**C**) almost completely after four days; x, no GSL-positive band; running solvent for neutral- and acidic GSLs was CHCl_3_/CH_3_OH, 0.2% CaCl_2_, 60:35:8 (*v*/*v*). (**D**,**E**, quantification) Miglustat treatment led to a visible arrest of the cell cycle in Lovo cells but not in HCT116 cells (**F**,**G**, quantification). (**H**–**K**) Miglustat inhibited tumor spheroid growth of Lovo- (**H**,**I**, quantification) and HCT116 cells (**J**,**K**, quantification); each data point on the graphs represents one biological replicate; scale bars, 100 µM; significances are *, *p* ≤ 0.05; **, *p* < 0.01; ***, *p* < 0.001.

**Figure 4 ijms-22-10539-f004:**
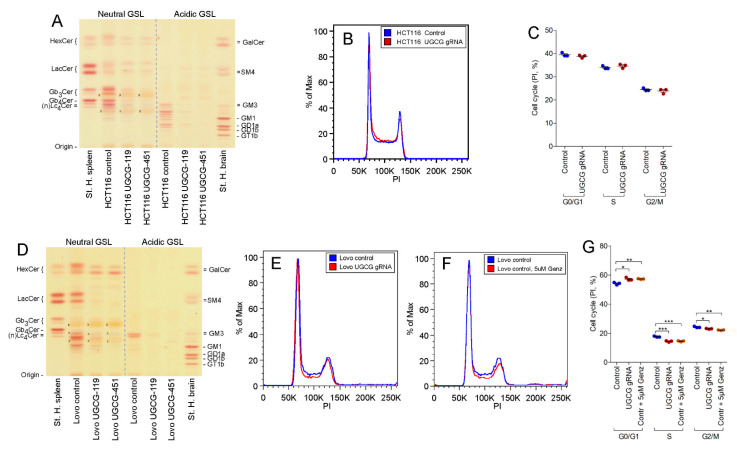
TLC and cell cycle determination of *UGCG*-gRNA treated HCT116- and Lovo cells. (**A**,**D**) An almost complete depletion of neutral and acidic GSLs was achieved in HCT116- (**A**) and Lovo cells (**D**) by *UGCG*-guide-RNA (gRNA); x, no GSL-positive band; running solvent for neutral- and acidic GSLs was CHCl_3_/CH_3_OH, 0.2% CaCl_2_, 60:35:8 (*v*/*v*). (**B**,**C**, quantification) *UGCG*-gRNA treated HCT116 cells did not show an arrest of the cell cycle, similar as HCT116 cells treated with low doses of Genz (Figure 2). (**E**–**G**, quantification) Genetic depletion using *UGCG*-gRNA-application (**E**) and treatment of Lovo cells with 5 µM Genz (**F**) resulted in an arrest of the cell cycle, whereas the response in Genz-treated cells was slightly more pronounced; significances are *, *p* ≤ 0.05; **, *p* < 0.01; ***, *p* < 0.001.

**Figure 5 ijms-22-10539-f005:**
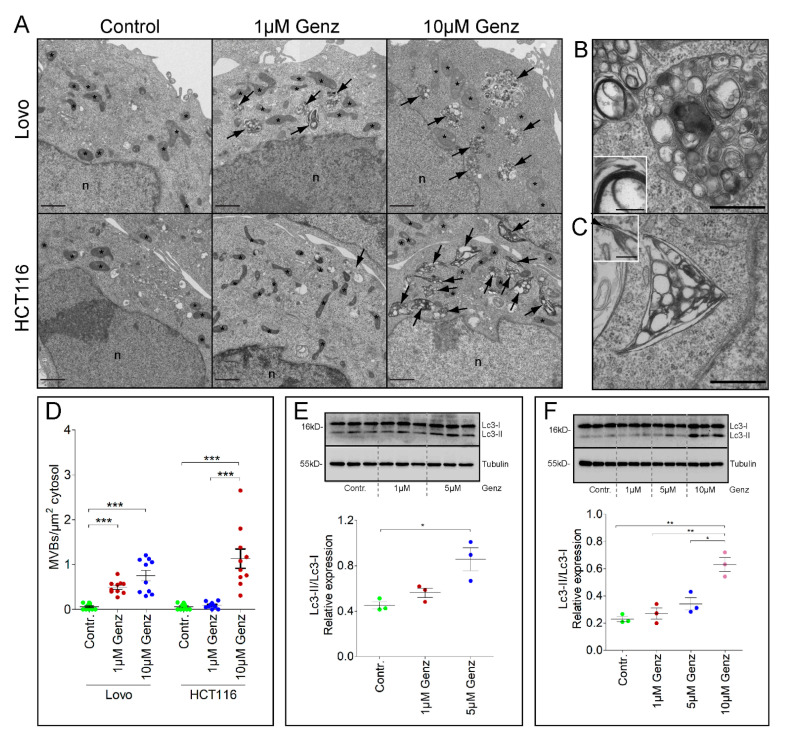
Lovo and HCT116 cells present an elevated number of multivesicular bodies (MVBs) which contain conspicuous structured vesicles upon Genz treatment. (**A**) The number of multivesicular bodies with vesicles displaying multilamellar structures was elevated in Genz-treated colon carcinoma cells. MVB-number increased markedly with 1 µM Genz in Lovo cells. Only low numbers of MVBs were seen in HCT116 cells upon treatment with 1 µM Genz. The numbers of MVBs rose markedly with higher Genz concentrations; scale bars, 1 µm. (**B**) Vesicles inside MVBs of Lovo cells were canonically spheroid, with potential to form onion-like superstructures and myelin-like compaction of stacked membranes (**B**, insert). (**C**) MVBs of HCT116 cells displayed angular overall shapes (here triangular), a combination of apparently stiff sheets with sharp cornered rims. The insert demonstrates a widening of the expectedly lipophilic middle-leaflet of the canonical bi-laminar unit-membrane (arrowhead), scale bars in (**B**,**D**), 500 nm; in insets 100 nm. (**D**) The number of MVBs increased with elevated Genz-concentrations; shown are mean numbers of MVBs per µm^2^ cytosol, each from 10 cells. (**E**,**F**) The expression levels of the autophagy marker Lc3-II rose markedly in both Lovo- (**E**) and HCT116 cells (**F**) upon increased Genz concentrations; significances are *, *p* ≤ 0.05; **, *p* < 0.01; ***, *p* < 0.001.

**Figure 6 ijms-22-10539-f006:**
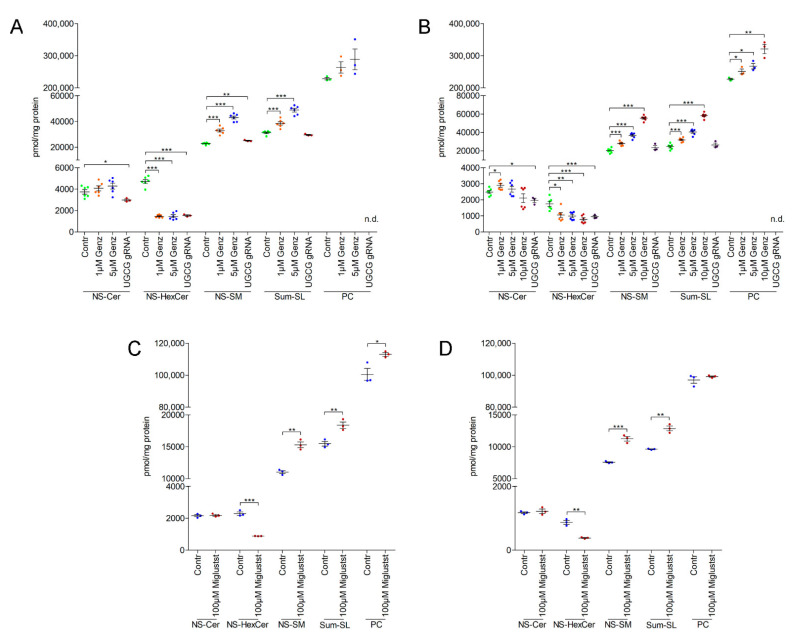
Genz treatment of Lovo and HCT116 cells lead to a concentration-dependent increase of sphingomyelin (SM). Lipids were quantified by C18-reversed phase UPLC/MS^2^ in MRM mode using internal standards for each lipid class. Major sphingolipids containing non-hydroxy fatty acids (N) and (C18)-sphingosine (S) were recorded; Cer, ceramide, HexCer: hexosylceramide, SM, sphingomyelin, and PC: phosphatidylcholine. (**A**,**B**) HexCer was reduced to a similar extent in Genz- or *UGCG*-gRNA-treated Lovo (**A**) or HCT116 cells (**B**). The SM content in Lovo (**A**) and HCT116 (**B**) cells was remarkably elevated upon Genz treatment and much higher as in cells treated with *UGCG*-gRNA in which the SM-increase correlated vice versa with the reduction of GSLs. The ceramide levels in Lovo and HCT116 cells upon Genz treatment were similar as in controls. (**C**,**D**) SM levels in Miglustat-treated Lovo (**C**) and HCT116 cells (**D**) were also elevated as compared to controls. Ceramides appeared unaltered upon Miglustat treatment; each data point on the graphs represents one biological replicate; significances are *, *p* ≤ 0.05; **, *p* < 0.01; ***, *p* < 0.001.

**Figure 7 ijms-22-10539-f007:**
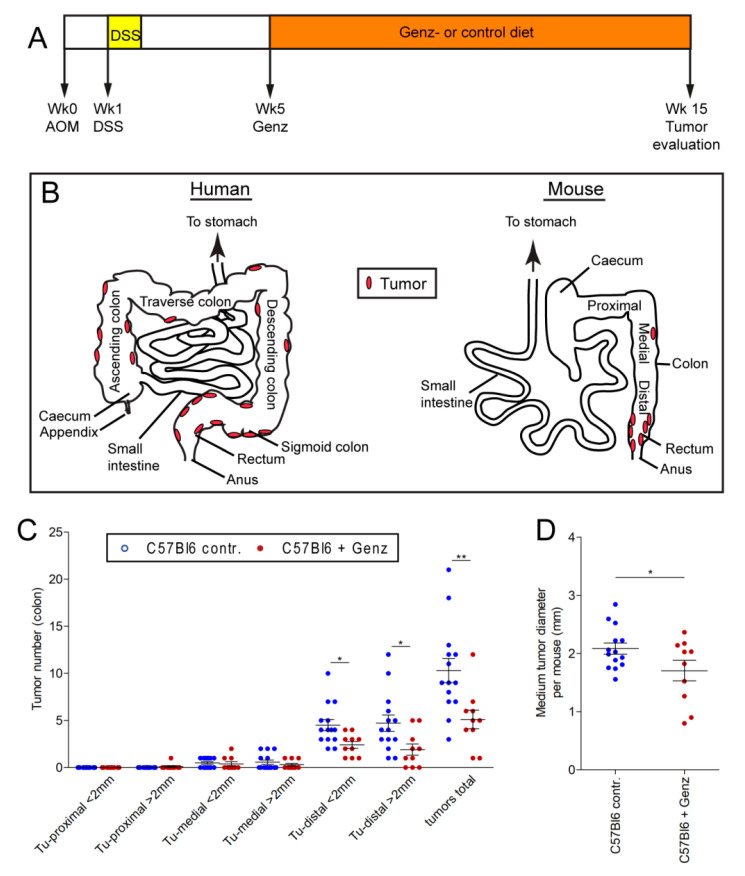
Genz treatment lowered the incidence of experimental CRC in vivo. (**A**) Scheme of the treatment course. CRC was induced in C57Bl6 mice with azoxymethane (AOM) at the age of 8 weeks, followed by a tumor promoting 5-day cycle of dextrane sulfate (DSS) applied by the drinking water starting one week after the AOM-treatment. Mice, *n* = 24, received chow diet until five weeks after tumor induction. Then one cohort of animals, *n* = 10, received 0.225% Genz in the food. All animals were sacrificed 15 weeks after tumor induction according to the termination criteria from the animal application. (**B**) Scheme of the human and murine intestinal tract. Colorectal tumors in humans develop preferentially in the areas of the sigmoid colon and the rectum but also in other areas of the large intestine. In this mouse model, we observed tumors predominantly restricted to the rectal area of the colon with a critical load at fifteen weeks after tumor induction. (**C**) Treatment of mice with Genz resulted in a markedly lower number of colorectal tumors in the distal part of the colon. (**D**) The size of tumors detected in the colons of Genz-treated mice was also notably smaller as compared to chow fed mice; each data point represents the mean size of all tumors detected in one mouse; significances are *, *p* ≤ 0.05; **, *p* < 0.01.

**Figure 8 ijms-22-10539-f008:**
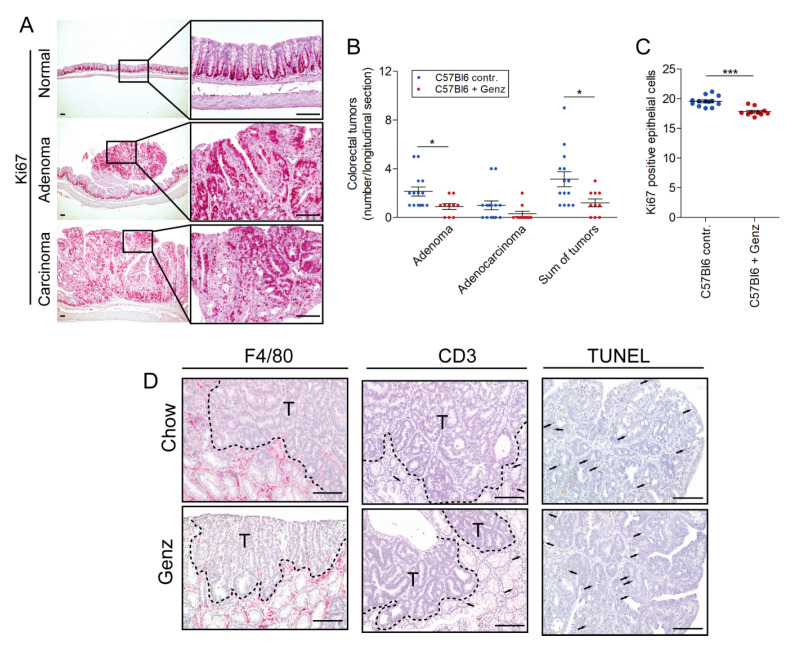
The number of colorectal adenomas were reduced in Genz-treated mice. (**A**) Ki67 staining indicated proliferative cells in colons of mice. Both Genz-treated mice and controls developed predominantly adenomas with its typical polypoid-like shape (**A**,**B**, middle images) and only low numbers of adenocarcinomas (**A**,**B,** lower images). However, Genz-feeding markedly reduced the number of adenomas as compared to chow-fed controls (**B**). (**C**) The number of Ki67 positive cells in the colorectal parts of the colons was lower in crypts of Genz-treated mice as compared to controls; each data point represents mean numbers of Ki67 positive cells of approximately 50 crypts per mouse. Counted were positive cells only from completely vertically cut villi apart from the tumor areas; significances are *, *p* ≤ 0.05; ***, *p* < 0.001. (**D**, F4/80) Both Genz-fed mice and controls lacked macrophage infiltrations in the tumor but showed an enrichment in peritumoral areas. (**D**, CD3) Only a few cells stained positive with the T cell marker CD3; T, tumor area. Genz treatment did not lead to increased apoptosis (**D**, TUNEL); a few cells with smaller nuclei than epithelial cells stained positive in tumors of treated mice and controls; scale bars, 100 µM.

**Figure 9 ijms-22-10539-f009:**
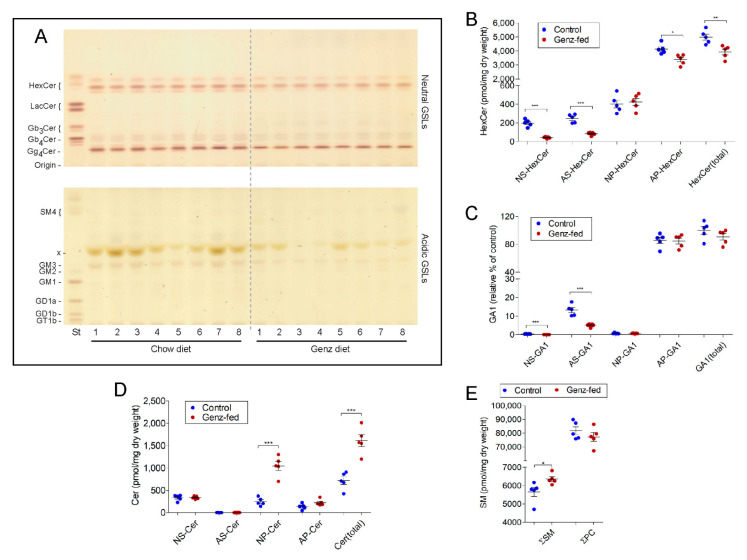
Genz treatment alters the sphingolipid composition in the intestine. (**A**) TLC-analysis of neutral and acidic GSLs from the intestine of Genz-treated mice. The intestine of adult mice contained predominantly the neutral GSLs HexCer and GA1. Lipids were quantified by C18-reversed phase UPLC/MS^2^ in MRM mode using internal standards for each lipid class except for GA1, which was normalized to internal HexCer standards. Sphingolipids containing either non-hydroxy fatty acids (N) or alpha-hydroxy fatty acids (**A**) in combination with (C18)-sphingosine (S) or with (C18)-phytosphingosine (P) were recorded; Cer: ceramide, HexCer, hexosylceramide, GA1: asialo-ganglioside 1 or gangliotetraosylceramide (Gg_4_Cer, see Figure 1A), and SM: sphingomyelin. (**B**,**C**) MS^2^-analysis of HexCer (**B**) and GA1 (**C**). AS- as well as NS-species markedly decreased in the intestine of Genz-treated mice. AP-GSLs were less effectively downregulated. Genz-application did not lead to a reduction of NP-GSLs; for designation of ceramide anchors, see abbreviations. (**D**) NP-Cers increased in intestines of Genz-treated mice as compared to chow-fed controls. (**E**) The SM content was markedly elevated and PC remained essentially unaltered upon Genz treatment; significances are *, *p* ≤ 0.05; **, *p* < 0.01; ***, *p* < 0.001.

## Data Availability

All data are available within the main body of the manuscript or in the Appendix A.

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
