# Peer review of "Blockade of Glycosphingolipid Synthesis Inhibits Cell Cycle and Spheroid Growth of Colon Cancer Cells In Vitro and Experimental Colon Cancer Incidence In Vivo"

_ijms, 2021, doi:10.3390/ijms221910539_

Round 1

Reviewer 1 Report

The manuscript demonstrated that GSL synthesis inhibitors Genz and Miglustat, and Cas9-based knockout of UGCG can reduce the proliferation of CRC cell lines and tumor growth. Genz alters the sphingolipid metabolism in the specific cell lines to reduce the GSL content but increase other lipids including ceramide and SM, and MVB content in cells. Such alteration may happen by altered degradation of sphingolipids, thus inhibiting the cell growth. The manuscript is based on solid methods, sound experimental design, and excellent presentation of data. 

  1. I have a few minor comments that authors may hopefully address before publication. The abnormal changes of lipids by Genz and other intervention are clear in mass-spec and TLC data, but the electron microscopy images cannot show the MVB or abnormal structures actually contain those lipids.  Thus it is not correct to demonstrate that the accumulation of lipids happens upon the treatment. I hope the authors can provide enough data or explanations to address this point. 
  2. The abnormal MVB structures should be quantified and shown to be different from controls. At least, independent experimental results should be shown in the figure/supplementary figures. 
  3. Some charts are too small to read. Particular figure 5 should be re-sized. 

Author Response

We thank reviewer 1 for the valuable remarks improving the impact of the manuscript.

  1. I have a few minor comments that authors may hopefully address before publication. The abnormal changes of lipids by Genz and other intervention are clear in mass-spec and TLC data, but the electron microscopy images cannot show the MVB or abnormal structures actually contain those lipids.  Thus it is not correct to demonstrate that the accumulation of lipids happens upon the treatment. I hope the authors can provide enough data or explanations to address this point. 

The lectin lysenin has been shown to bind specifically to sphingomyelin. We intended to use alexafluor-labeled lysenin to show intracellular accumulation of sphingomyelin. Unfortunately, the lysenin from the only available source was of bad quality; it was clumpy and appeared insoluble in the labeling buffer. Nevertheless, we continued labeling but lost the conjugate completely during the purification procedure and could not perform the staining.

We have solid indirect evidence of lipid accumulation in vesicles after Genz treatment.

  • By EM, we found elevated numbers of MVBs displaying onion-like membrane structures after Genz treatment. An accumulation of membranes is equivalent to increased lipid content, which correlates also well with the intensive EM staining. Similar ultra-structures as seen in the colon cancer cells after Genz treatment can be found in cells of lipid storage disease patients such as Gaucher, Niemann-Pick, or Tay-Sachs.
  • Cationic amphiphilic drugs (CADs) inhibit the catalytic degradation of sphingomyelin by sphingomyelinases in lysosomes. GCS-inhibitors such as Genz fulfill the property of a CAD and thus sphingomyelin storage in vesicular compartments appears very likely.
  • Recently, a manuscript published by Hartwig et al. 2021 Jun 30;22(13):7065. doi: 10.3390/ijms22137065 clearly has shown that PDMP, a GCS-inhibitor structurally related to Genz, caused accumulation of sphingomyelin in lysosomal compartments. The results of this group were already discussed in the initial submitted version. But since there might be a close relation to our observations, we referred to this publication now also to the ‘Results’ section.
  1. The abnormal MVB structures should be quantified and shown to be different from controls. At least, independent experimental results should be shown in the figure/supplementary figures.

As recommended, we have quantified the number of MVBs in controls, and Genz-treated Lovo and HCT116 cells and added the results to the results section and to Figure 5, which has been completely rearranged.

  1. Some charts are too small to read. Particular figure 5 should be re-sized. 

As suggested, the size of the fonts of the graphs in Figure 5 has been increased. 

Reviewer 2 Report

The manuscript entitled “ Blockade of glycosphingolipid synthesis inhibits cell cycle and pheroid growth of colon cancer cells in vitro and experimental olon cancer incidence in vivo” shows that glycosphingolipid (GSL)-depletion by inhibition of glucosylceramide synthase, the major enzyme involved in the initial step of GSL-bio-synthesis, led to growth retardation of colon cancer cell-derived tumor microspheres and o an arrest of the cell cycle. In a mouse model mimicking human CRC, marked reduction particularly of colorectal adenomas was achieved by treatment of animals with the GCS-inhibitor Genz-123346. The exact molecular mechanism of the Genz-dependent decrease of the tumor incidence in the CRC-model needs to be further elucidated. However, since inhibitors of GSL-biosynthesis are clinically available, this study may have a direct translational implication as a potential therapeutic application for patients suffering from colorectal cancer.

English language and style check are required.

Author Response

As suggested by reviewer 2, the manuscript has been proof-read by a native English speaking colleague and the English language and style improved.